# Retrieving high-resolution information from disordered 2D crystals by single-particle cryo-EM

Ricardo D. Righetto[1], Nikhil Biyani[1], Julia Kowal[1,2], Mohamed Chami[1] & Henning Stahlberg [1]

Electron crystallography can reveal the structure of membrane proteins within 2D crystals under close-to-native conditions. High-resolution structural information can only be reached if crystals are perfectly flat and highly ordered. In practice, such crystals are difficult to obtain. Available image unbending algorithms correct for disorder, but only perform well on images of non-tilted, flat crystals, while out-of-plane distortions are not addressed. Here, we present an approach that employs single-particle refinement procedures to locally unbend crystals in 3D. With this method, density maps of the MloK1 potassium channel with a resolution of 4 Å were obtained from images of 2D crystals that do not diffract beyond 10 Å. Furthermore, 3D classification allowed multiple structures to be resolved, revealing a series of MloK1 conformations within a single 2D crystal. This conformational heterogeneity explains the poor diffraction observed and is related to channel function. The approach is implemented in the FOCUS package.

[1] Center for Cellular Imaging and NanoAnalytics, Biozentrum, University of Basel, Mattenstrasse 26, CH-4058 Basel, Switzerland. [2] Institute for Molecular Biology and Biophysics, ETH Zürich, Otto-Stern-Weg 5, CH-8093 Zürich, Switzerland. Correspondence and requests for materials should be addressed to H.S. (email: henning.stahlberg@unibas.ch)

Electron crystallography of native two-dimensional (2D) crystals of bacteriorhodopsin allowed the determination of the first 3D model of a membrane protein in 1975[1]. Since then, considerable effort has been invested in growing 2D crystals of purified membrane proteins from protein-lipid-detergent mixtures, leading to several high-resolution structures[2–5]. However, in most cases, grown 2D crystals only diffracted to lower resolution[6]. Electron crystallography also requires collecting image data from tilted samples, which is technically difficult and limited in the reachable tilt angle, causing the so-called "missing cone" problem in Fourier space[7]. Furthermore, images acquired from tilted samples are of lower quality when compared to images of non-tilted samples, because the increased effective specimen thickness also increases the number of inelastic electron scattering events and reduces the image contrast[8]. These effects combined limit the resolution along the vertical z-direction, which may make reconstructions appear vertically smeared-out in real space.

Recently, the resolution revolution[9] in cryo-electron microscopy (cryo-EM), triggered by the development of direct electron detectors (DED)[10] and better image processing software, allowed determining the atomic structures of isolated membrane protein particles in detergent or amphipols[11], or lipid nanodiscs[12]. In particular, DEDs deliver images at much improved signal-to-noise ratios (SNR) and allow series of dose-fractionated images (movies) to be recorded from the same region, which can be computationally corrected for image drift and merged. Single Particle Analysis (SPA) is now a widespread method capable of determining high-resolution protein structures routinely.

Nevertheless, the capability to analyze the structure of membrane proteins in 2D crystals is important, when (i) such crystals occur naturally in the cell membrane, (ii) the lipid bilayer influences membrane protein function, or (iii) the protein of interest is too small for conventional SPA and the addition of tags to increase the particle size would disturb its function. 2D crystals can also help elucidating conformational changes triggered by ligands[6,7]. In addition, recent advances toward the rational design of scaffolds may provide a more systematic way to present arbitrary proteins as 2D crystals for structural studies[13,14]. The capability to reach highest-resolution structural data from badly ordered 2D crystals is important.

In electron crystallography, distortions of the 2D crystal lattice in the image plane can be computationally corrected via an interpolation scheme[15], correlation averaging[16], or the so-called lattice "unbending" algorithm[17]. However, this unbending is performed in the 2D projection images only. Three-dimensional (3D) out-of-plane distortions in the crystals, i.e., curvature or "bumps" in the membrane plane, could not be corrected. SPA, on the other hand, aligns projections of randomly oriented isolated particles in 3D space to reconstruct the density map of the underlying protein structure[18,19]. Thus, if the unit cells or patches of the 2D crystals are treated as "single particles", SPA offers the framework required to correct for out-of-plane crystal distortions. This rationale is similar to that of processing segments extracted from helical filaments[20–22]. Previous attempts of "3D unbending" 2D crystal datasets did not reach higher resolutions than the conventional 2D crystallographic approach, despite exploiting natural constraints on the orientation of particles extracted from 2D crystals[23,24]. The lack of DED data at the time prohibited correction for specimen drift, and the algorithms employed were suboptimal compared to the modern, probability-based methodology now used in SPA to account for noisy data while avoiding reference bias[25–28].

Here, we present a high-resolution application of SPA to electron crystallography data, using cryo-EM movies of the prokaryotic, cyclic-nucleotide modulated potassium channel MloK1. Previous studies by crystallographic processing on this dataset led to the structure of MloK1 at 4.5 Å[29]. Each of the four MloK1 monomers forming the pore has a transmembrane domain (TMD), a voltage sensor domain (VSD), and a soluble cyclic-nucleotide binding domain (CNBD), which lies in the intracellular side. The total molecular weight of the tetramer is 160 kDa. Although the 2D crystal images processed did not diffract beyond 10 Å, we improved the resolution of the MloK1 3D map to 4.0 Å. Furthermore, we identified different conformations of MloK1 tetramers within the disordered 2D crystals by means of single-particle classification.

## Results

**Software implementation**. As an extension to recently developed movie-mode electron crystallography algorithms[8], we implemented a 2D crystal single-particle module into the FOCUS software package[30] (Fig. 1). Electron dose-fractionated movies of 2D crystals are first corrected for specimen drift and averaged within FOCUS, using external tools[31,32]. Subsequently, microscope defocus, sample tilt geometry, crystal lattice vectors, and unit cell positions are determined for all movies[8,33]. This can be done in an automated and parallelized manner in FOCUS[30,33–35]. Next, the graphical user interface (GUI) features a new tab called "Particles" in which the user can perform particle picking, i.e., window patches from the 2D crystal images (Suppl. Fig. 1), which are boxed, assembled into a particle stack, and submitted towards implemented SPA workflows, using RELION[27] or FREALIGN[28] within FOCUS. If more than one lattice is present in one image, only the strongest lattice is considered for picking. The center of each windowed patch, here termed "particle", corresponds to the center of a crystal unit cell located by the classical unbending algorithm[17,36], optionally with an additional phase shift applied to translate the center of a protein to the center of the window (Suppl. Fig. 2). Only particle positions are considered, for which the cross-correlation (CC) peaks found by the unbending algorithm are stronger than a user-definable threshold. Overlap between boxes containing neighboring particles ensures smooth transitions of the local alignment parameters across the distorted 2D crystal lattice.

Alongside the particles, a meta-data file is generated containing information for every particle, such as the micrograph it came from, its Euler angles converted from the 2D crystal tilt geometry[8], the picking x,y coordinates, and the calculated defocus and astigmatism values at the center of the box following the CTFTILT conventions[37]. Optionally, an additional particle stack can be created simultaneously during picking by correcting each particle box for the local contrast transfer function (CTF) of the electron microscope. Available CTF correction methods are phase-flipping, CTF multiplication, or an ad hoc Wiener filter. When generated, the CTF-corrected stack can be used to perform correlation averaging[16] of the 2D crystals, providing immediate feedback on the dataset quality in similar fashion to the 2D class averages generated in SPA (Suppl. Fig. 3). Since 2D crystal images present the proteins densely packed in the images, only a comparatively small number (100–1000) of 2D crystal images is usually needed to produce large particle numbers (100,000 to 1,000,000). Windowed particles for each 2D crystal image can be averaged, and these fewer image averages can already be used to determine initial 3D models and pre-refine the alignment parameters that will be propagated to the larger particle dataset, thereby speeding up the structure determination process (Fig. 1).

**Processing MloK1 2D crystals**. In order to test our approach, we utilized a 2D crystal dataset of the potassium channel MloK1 in the presence of cAMP that yielded a 4.5 Å resolution map when processed by classical 2D crystallography[29]. The Fourier

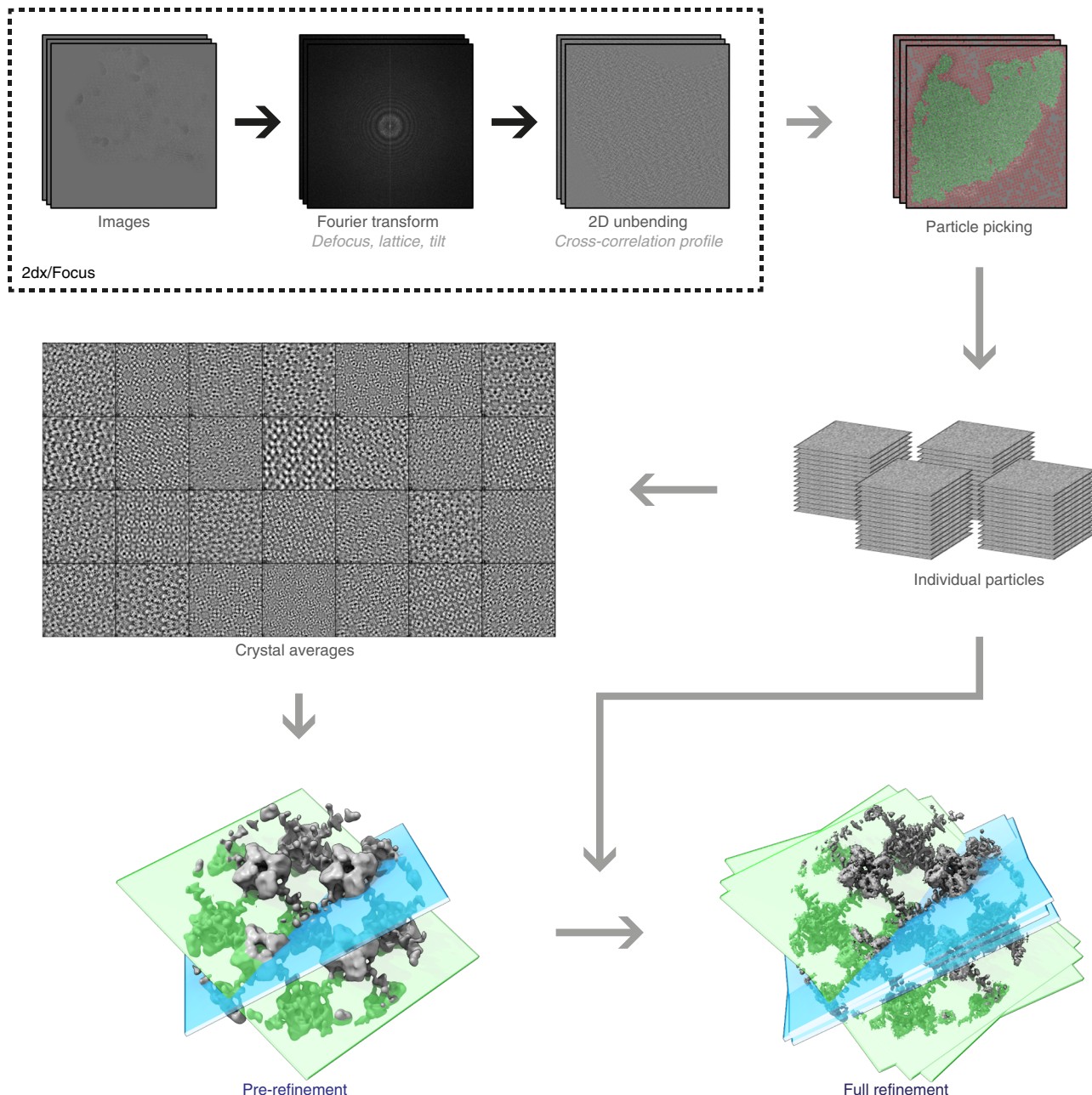

**Fig. 1** Workflow employed to process 2D crystal data by single-particle analysis programs. Steps depicted within the dashed box were previously available in the 2D crystallography mode of FOCUS. A new GUI (Suppl. Fig. 1) was implemented to pick particles from the 2D crystals based on the unit cell positions obtained from classical unbending. Stacks of individual particles are then exported together with the associated metadata for processing with established SPA software. The particles extracted from the 2D crystal in each image are averaged in FOCUS for quality assessment, and rapid pre-refinement of the map is done with FREALIGN using these crystal averages as input. For simplicity, slices representing two crystal averages in different orientations are shown (light blue and light green). Then, the individual particles are initialized with the pre-refined alignment parameters and provided to FREALIGN to perform a full refinement to high resolution. These individual particles might end up having slightly different orientations than the crystal average, due to distortions in the 2D crystal lattice

transforms of the images showed Bragg reflections to 10 Å resolution at best, in most cases worse than 14 Å. Drift-correction of the 346 movies from the previous study was performed with MotionCor2[32] in FOCUS[30] and the aligned averages were used for further processing. The defocus at the center of the image, tilt geometry and lattice determined previously were retained. Out of the 346 crystal images, 76 were discarded, as they were associated with a second lattice in the same image. The image data from secondary lattices can be treated independently in classical 2D crystallography, but in SPA processing these would lead to

exaggerated resolution estimates due to overlap between particles picked from different lattices in the same image, i.e., it would effectively introduce duplicated particles in the dataset. This can occur for example if two 2D crystals happen to be on top of each other on the support film, or if the periodical structure under study has a 3D component, such as a microtubule or vesicle, which becomes flattened during grid preparation[7]. Using the new GUI, a total of 231,688 unique particles with a box size of 320 × 320 pixels were windowed from the remaining 270 unique images, which had a pixel size of 1.3 Å on the sample level (Suppl.

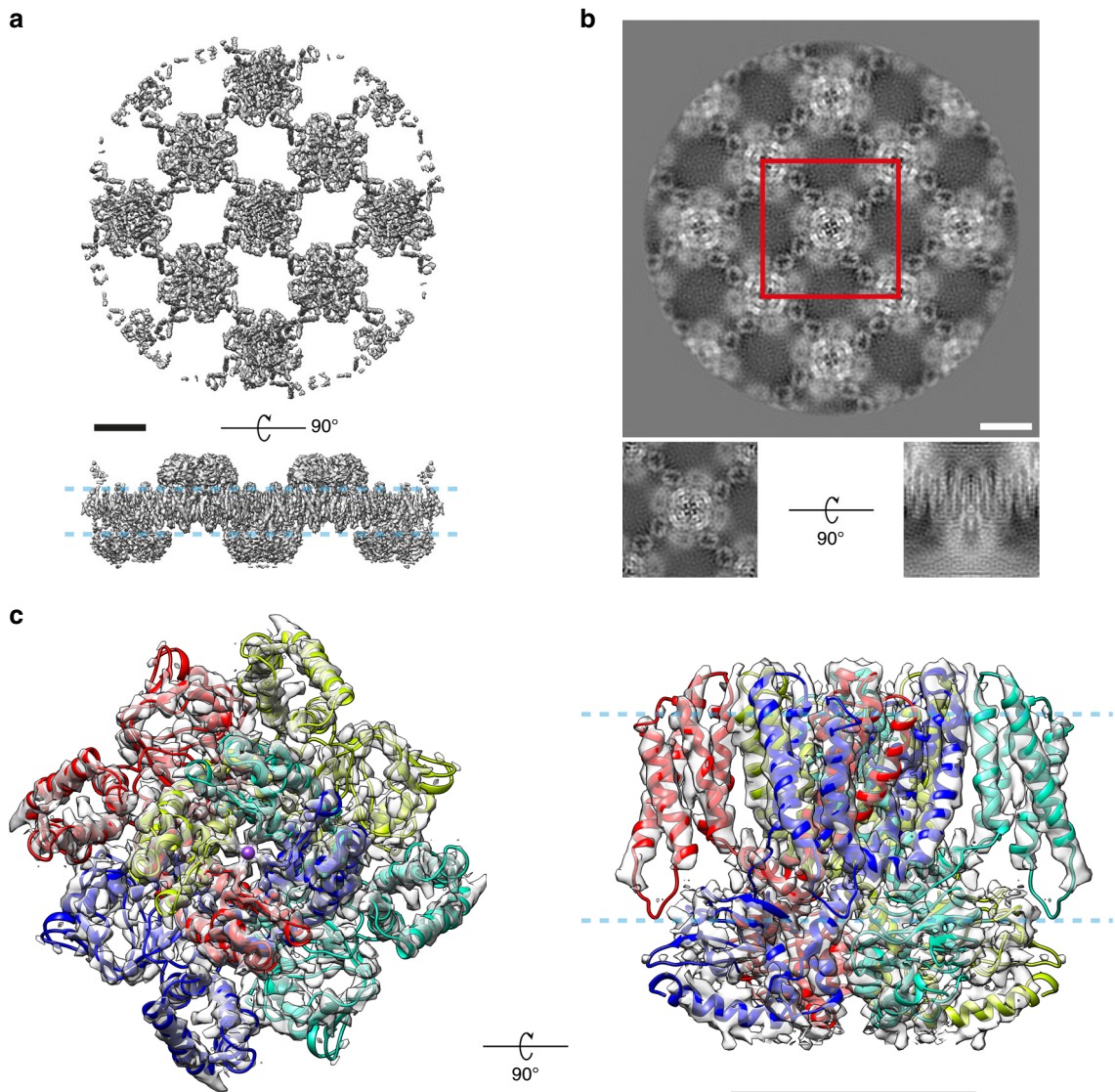

**Fig. 2** The cryo-EM map of full-length MloK1 at 4 Å. The map was obtained from the consensus single-particle refinement of 2D crystal data in FREALIGN. **a** The full refined map containing approximately nine MloK1 tetramers; total molecular weight over 1 MDa. **b** Projection of the full map along the z axis. The sub-volume used for postprocessing is indicated (red box). Insets: projections of the central sub-volume orthogonal to the xy and xz planes. **c** The postprocessed map of the central tetramer with the refined atomic model fit into the density, viewed from the extracellular side and parallel to the membrane plane. The pale blue dashed lines indicate the approximate position of the lipid bilayer. Scale bars: 50 Å

Fig. 2). In the non-tilted views, each "particle" was roughly comprised of nine MloK1 tetramers. Because the unit cell of the processed MloK1 2D crystals had $P42_12$ space-group symmetry that contains a screw axis[29], we applied a phase origin shift of half a unit cell (180°) in the direction of the first lattice vector to the picking coordinates in order to have one tetramer at the center of each particle box. Phase-flipped copies of the particle projections were calculated and stored at the same time as the non-CTF corrected particle projections for the generation of crystal averages (Suppl. Fig. 3).

**Consensus refinement.** Using a modified version of FREALIGN v9.11[28] (Suppl. Note 1) and the initial tilt geometry obtained in FOCUS, we calculated a 3D reconstruction at 6.5 Å resolution with C4 symmetry imposed. After pre-refinement using the 270 crystal averages, the global resolution improved to 4.8 Å. Finally, the updated alignment parameters were propagated from the crystal averages to the 231,688 individual raw particles, and refined in a single class using a custom auto-refinement script

written for FREALIGN (Suppl. Note 2). The refined map contains nine full MloK1 tetramers (Fig. 2a). For further analysis, the central tetramer was cropped out of the full reconstruction and postprocessed (Fig. 2b) leading to the "consensus" refinement map at a global resolution of 4.0 Å based on the Fourier shell correlation (FSC) curve[38–40] (Fig. 2c and Suppl. Fig. 4). To avoid inflation of the FSC curve due to the large overlap between neighboring particle boxes, particles extracted from the same 2D crystal were always assigned to the same half-set throughout the refinement[21]. At this resolution level we could identify densities for many side chains in the transmembrane domains (TMD), especially the larger ones, such as phenylalanine and tryptophan and those at the S4–S5 linker (Suppl. Fig. 5). This 4.0 Å resolution estimate based on the FSC corresponds to an isotropically averaged measurement. The map's resolution is better in the xy membrane plane than it is along the z axis. By calculating the Fourier ring correlation between the central xy slices ($z = 0$) of the unmasked half-maps we estimate resolution to be slightly better than 4.0 Å in this plane, whereas in the orthogonal xz plane

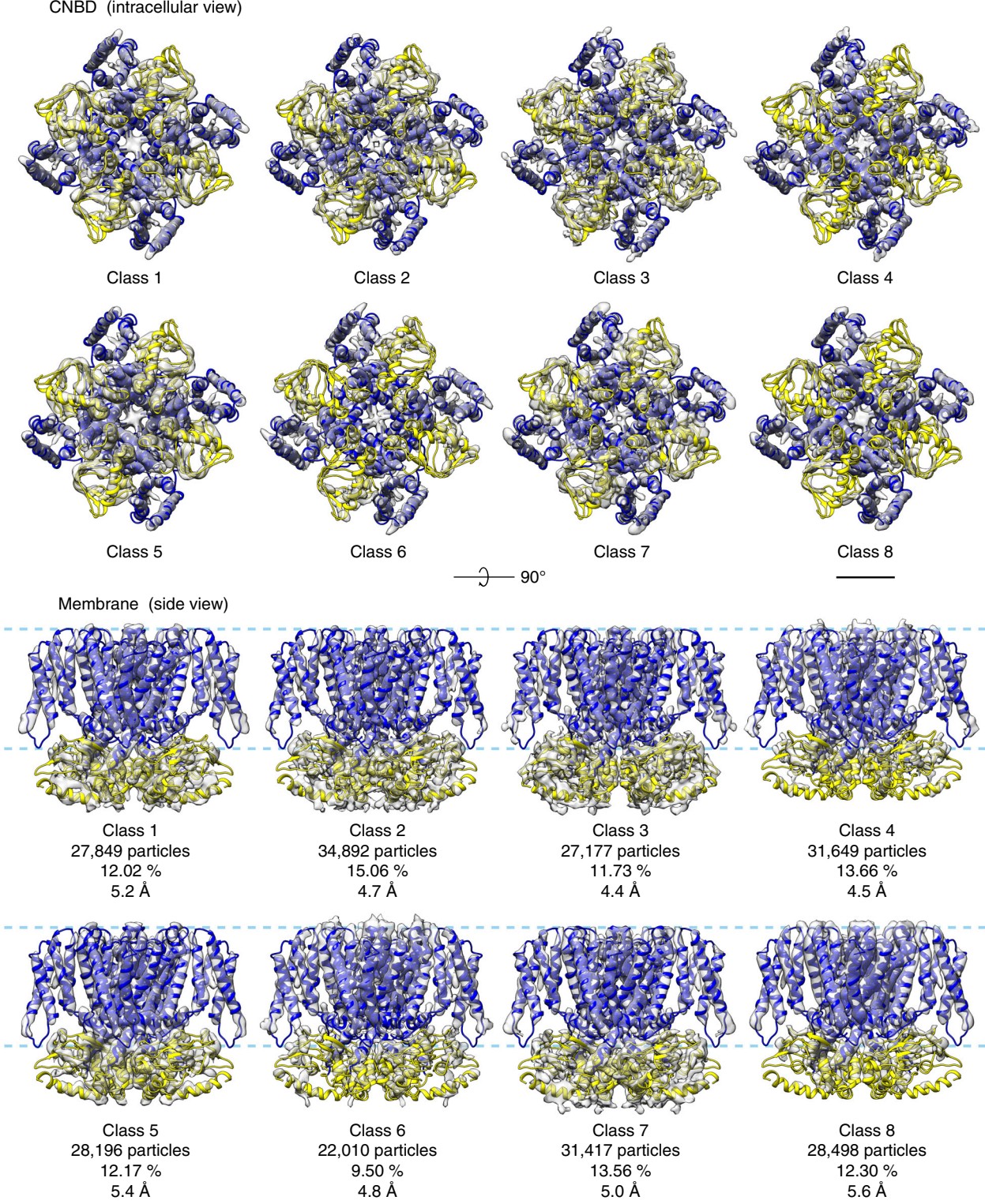

**Fig. 3** Conformational continuum of MloK1 from 2D crystals. All the maps were obtained from one 2D crystal dataset using signal subtraction and 3D classification after consensus refinement. The consensus model was flexibly fitted and refined into each 3D class. The CNBD is colored yellow and the TMD is colored blue, while the maps are shown as transparent gray surfaces. The map threshold for each class was calculated such that all iso-surfaces enclose the same volume. The pale blue dashed line indicates the approximate position of the lipid bilayer. Conformational variations among the classes primarily change the height of the CNBDs in relation to the membrane slab and their tilt in relation to the TMD and the VSD. C4 symmetry was applied. Scale bar: 25 Å

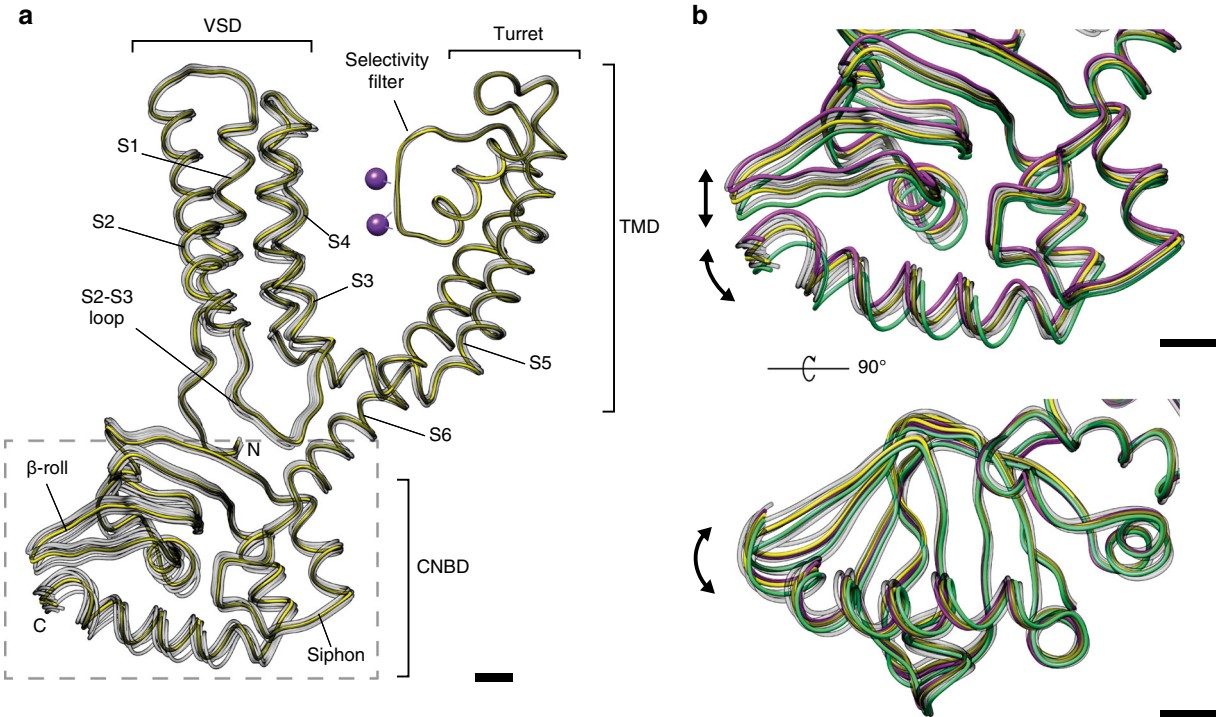

**Fig. 4** Conformational ensemble of MloK1 within the 2D crystals. The consensus model (yellow) and the eight atomic models derived from the 3D classes (gray) are shown superposed. For this visualization, the eight models in the ensemble were aligned to the selectivity filter of the consensus model (residues 174–180). **a** One full chain of the MloK1 tetramer is shown with the different domains indicated; the dashed light-gray box indicates the area highlighted for panels **b** and **c**; **b** close-up view of the CNBD part where most conformational changes occur. Models #1 and #4 are the most different from each other (RMSD 1.021 Å) and are depicted in green and magenta, respectively. The arrows indicate the principal directions of conformational variability. Scale bars: 5 Å

($y = 0$), which is identical to the $yz$ plane due to the imposed symmetry, the estimated resolution is about 4.7 Å considering the FSC 0.143 threshold[39] (Suppl. Fig. 4f). These two orthogonal resolution estimates are consistent with the observed features of the consensus map. Rigid-body fitting of our previously published model for MloK1[29] into the higher-resolution consensus map revealed a sequence register shift of 1 residue at the S4–S5 linker, while the X-ray crystallographic model for the TMD[41] (PDB: 3BEH) was in good agreement with the experimental densities of our map. This new map allowed refining a full-length atomic model of MloK1, improving both its geometry and fit-to-density indicators compared to the previous model (PDB: 6EO1) obtained by classical 2D electron crystallographic processing[29] (Suppl. Table 1).

**3D Classification**. The local resolution estimates and model B-factors for the consensus map (Suppl. Fig. 4a, b) suggested high conformational variability of the CNBDs of the molecules, inviting for 3D classification of the particles. Because particle images contained more than one MloK1 tetramer, signal subtraction[42] for the densities corresponding to the neighboring tetramers was applied (Suppl. Fig. 6), in order to be able to perform a 3D classification on the central tetramer alone. This was especially important for images of tilted samples, where the projections of neighboring tetramers partially overlap with the projection of the central tetramer, which is our classification target. We then applied maximum-likelihood 3D classification in FREALIGN[28] to determine conformational variations within the 2D crystals, while imposing C4 symmetry and keeping the alignment parameters for the particles constant. Surprisingly, we found that the dataset was quite heterogeneous. 3D classification produced eight 3D classes at resolutions from 4.4 to 5.6 Å (Fig. 3

and Suppl. Fig. 7) that differed conformationally to various degrees (see below for more details). The 3D classification did not correlate with particle positions in the 2D crystals, indicating that no bias from defocus or tilt angle affected the classification (Suppl. Fig. 8). When employing a localized reconstruction method[43,44] to look for non-symmetric conformations, no significant deviations from four-fold symmetry were detected.

To obtain more insight, the consensus atomic model was flexibly fit into each map of the 3D classes using Normal Mode Analysis[45] and then refined in real space[46]. The resulting models were globally aligned against the consensus map (Suppl. Fig. 9), showing root mean square deviations (RMSD) ranging from 0.379 Å (class 7) to 0.720 Å (class 1) toward the consensus map. Alignment of the models against the consensus model by the selectivity filter region only (residues 174–180) produced an ensemble depicting a continuum of conformations of the CNBDs and the S2–S3 loops (Fig. 4), within which model #1 is the most different from all other models (Suppl. Fig. 9b) and in an "extended" conformation, with the CNBD farthest away from the membrane plane and the TMD. Conversely, model #4 is the most different from model #1 (RMSD of 1.021 Å after global superposition) and is in a "compact" conformation having the CNBD closest to the membrane plane and the TMD (Fig. 4, Suppl. Fig. 9). The other five models can be understood as intermediate snapshots along the trajectory between models #1 and #4.

Ranking the models in our ensemble in descending order of their pairwise RMSD values (Suppl. Table 2 and Suppl. Fig. 9a) allowed us to inspect the largest conformational changes of MloK1. Along the trajectory from the most compact state (model #4), to the most extended state (model #1) depicted in Suppl. Movie 1, the CNBD moves toward the inner pore axis of the channel and simultaneously away from membrane plane. The

C-terminal helix of the CNBDs tilts by about 3 degrees further from the membrane, while shifting away from the pore axis. The S2–S3 loop of one monomer closely follows the CNBD β-rolls of the adjacent monomer, which move toward the symmetry axis. At the same time, helix S4 of the VSD extends toward the intracellular membrane boundary. The next largest conformational difference appears between the "extended" model #6 and the "compact" model #1 (RMSD: 0.927 Å, Suppl. Table 2). This trajectory is very similar to that between models #4 and #1, but in addition the helices S1–S4 of the VSD tilt by about 1 degree in such a way that the S1–S2 loop comes closer to the periplasmic pore turrets (Suppl. Movie 2). Among the largest conformational changes observed is a 6-degree helical rotation of the CNBD around the symmetry axis, perpendicular to the membrane plane, between intermediate models #3 and #5 (RMSD: 0.845 Å, Suppl. Table 2), as shown in Suppl. Movie 3. Compared to model #1, the C-terminal helix in "extended"-like model #3 is rotated by about 4 degrees clockwise when seen from the CNBD side, while in the "compact" model #5 it is rotated by about 3 degrees counter-clockwise. A rotation of the selectivity filter by about the same magnitude also occurs in this trajectory. Finally, the most extended conformers of our ensemble (models #1 and #3) also have the S6 helix crossing bundle and the selectivity filter slightly more constricted (<1 Å) than the most compact ensemble members (models #4, #5, and #6).

## Discussion

Electron crystallography applied to well-ordered 2D crystals has, in the past, delivered structures of membrane proteins at far better resolutions than achievable by SPA. However, the resolution revolution in cryo-EM reversed this, so that today the precision of the physical arrangement of proteins in the 2D crystals is lower than the resolution achievable by SPA. With the addition of detergents or reconstitution in lipid nanodiscs, ion channels similar to MloK1 in mass have been consistently resolved in the resolution range of ~3.5 Å by SPA recently[12,47–49]. Nevertheless, cases exist where the structure of a membrane protein in the lipid environment provided by a 2D crystal is important, such as when the crystals occur naturally in the cell membrane, or when a 2D periodic array is artificially designed[13,14].

Here, we present a hybrid approach for the analysis of 2D crystal images, combining 2D crystallographic and single-particle image processing methods. Even though the missing cone effect is still present, and images of tilted samples are generally of lower quality than of non-tilted samples, the application of SPA algorithms to the MloK1 dataset ameliorated the overall resolution by accounting for local variations in the tilt geometries along the distorted 2D crystals, thus offering a higher diversity of views for the 3D reconstruction. This documents that the lack of crystal planarity can become an advantage in SPA. Also, masking the reference map along the iterative refinement in SPA is akin to projective constrained optimization[8,50], both methods contributing to reduce the impact of the missing cone.

Our highest resolution structure, derived from the consensus map (Fig. 2), is in general agreement with our previously published model[29]. The overall resolution of the map improved from 4.5 to 4.0 Å using our single-particle approach. The main bottlenecks to resolution by this method are presumably the lack of side views due to the tilting limitations, and the inherent lower quality of the high tilt images. Also, in this particular MloK1 dataset the pixel size at the sample level was relatively large (1.3 Å), which means the data was collected at a range of the K2 Summit's camera detective quantum efficiency that is suboptimal from the perspective of the SNR[10]. However, we here show that the previously determined structure was an average of several coexisting conformations. Consequently, SPA processing enabled the particles within the 2D crystals to be classified, revealing significant conformational variations. While 2D classification of unit cells has been explored in the past[51,52], our approach enables the retrieval of distinct 3D classes from within the same 2D crystal dataset. The eight obtained 3D classes (Fig. 3) have a slightly lower resolution than the consensus map, likely due to the limited number of particles in each class. The conformational differences observed between the individual 3D classes explain the low resolution to which the imaged 2D crystals showed diffraction. The analysis suggests that the CNBDs were only partially occupied with ligands in the cAMP-saturated crystals. Following Rangl et al.[53], a possible interpretation is that the extended "up-state" orientation of the CNBDs corresponds to the cAMP-free orientation, while the compact "down-state" orientation with the CNBDs approaching the membrane, is the cAMP-bound conformation[29]. Such insights have been obtained previously from lower resolution structures[54] and high-speed AFM experiments[53]. Our data and the processing workflow presented here, now provides higher-resolution data that invite for a more detailed analysis of this hypothesis in 3D models of MloK1.

Variations in CNBD positions are correlated with changes in the transmembrane regions including the VSD of MloK1, yielding insights into the mechanistic links between CNBDs and channel modulation. The observed interaction between the S2–S3 loop of the VSD and the CNBD β-rolls of the adjacent monomer corroborates the coupling between VSD and CNBD previously hypothesized[29]. In this proposed mechanism, the channel is open and in a compact conformation when cAMP is bound, and, upon ligand unbinding, the CNBD extends away from the membrane and closes the channel, accompanied by a tilting of the VSD towards the extracellular side (Suppl. Movie 2). This suggests that a change in the binding state of the CNBD is transduced to the selectivity filter via the VSD, possibly also with help of the N-terminal loop and the S4–S5 linker (Suppl. Movie 1 and Suppl. Movie 2), rather than directly via the C-linker of S6. Furthermore, we also observe a helical rotation of the CNBD around the pore axis during the CNBD extension, which is orchestrated with a rotation of the selectivity filter in the opposite direction (Suppl. Movie 3). Combined with a constriction of the pore, we suggest this mechanism to then prevent the passage of K+ ions at the selectivity filter[29,55,56]. While the S6 helix crossing bundle also constricts slightly during this conformational change, it remains too wide to block ion conduction.

In summary, we have shown that the SPA approach greatly reduces the need for perfectly planar, well-ordered 2D crystals. The procedure allows retrieving high-resolution information from disordered and non-flat 2D crystals, as illustrated by the 4 Å consensus model presented. Importantly, SPA also allows detecting conformational variations of proteins in the 2D crystals. Classification of the low-resolution 2D crystal images of the potassium channel MloK1 by SPA resulted in a series of 3D maps with resolutions between 4.4 and 5.6 Å, giving insight into ligand binding and channel gating. The data processing workflow is available from a GUI in the FOCUS package.

## Methods

**Protein expression, purification, and 2D crystallization**. The expression and purification of MloK1, and the growth of 2D crystals in the presence of cAMP is described in Kowal et al.[54]. For convenience, the protocol is summarized here: *E. coli* BL21(DE3) cells with the 6-His-tagged MloK1 construct were grown at 37 ℃. Protein expression was then induced with 0.2 mg/ml anhydrotetracycline for 2 h. Cells were sonicated and membrane proteins solubilized with 1.2% *n*-decyl-β-D-maltopyranosite (Anatrace) in the buffer containing 295 mM NaCl, 5 mM KCl, 20 mM Tris-HCl pH 8.0, 10% glycerol, 1 mM phenylmethylsulphonyl, and 0.2 mM cAMP, and incubated for 2.5 h at 4 ℃. The MloK1 extract was purified at a Co²⁺ affinity chromatography column, in the same buffer as before but with the addition

of 0.2% *n*-decyl-β-D-maltopyranoside and 40/500 mM imidazole (wash/elution). Throughout the purification, 0.2 mM cAMP (Fluka) was present to maintain the integrity of MloK1. For 2D crystallization, detergent-solubilized MloK1 sample was subsequently mixed with *E. coli* polar lipid extract (Avanti Polar Lipids) at a lipid-to-protein ratio of 0.8–1.0 (w/w) and dialyzed against detergent-free buffer (20 mM KCl, 20 mM Tris-HCl pH 7.6, 1 mM BaCl2, 1 mM EDTA, 0.2 mM cAMP) for 5–10 days.

**Sample preparation.** Grids for transmission electron microscopy were prepared as described in Kowal et al.[29]: 4 µl of the MloK1 2D crystal solution (0.8 mg/ml) were applied to glow-discharged Quantifoil holey carbon grids (R3.5/1, Cu 400 mesh), which had been coated with an additional <3-nm-thin amorphous carbon layer. Using an FEI Vitrobot IV with the environmental chamber set at 90% humidity and 20 °C, excess solution on the grid was blotted for 3.5 s, and the grids were then flash-frozen into liquid ethane.

**Cryo-EM imaging.** The dataset of 346 movies recorded and processed for Kowal et al.[29] was employed. As reported, the data were collected on an FEI Titan Krios TEM equipped with a Gatan K2 DED. Total dose: $40\,e^-/Å^2$ distributed over 40 movie frames. Pixel size: 1.3 Å on the sample level (counting mode). Nominal tilt range: −55° to +55º.

**Data processing.** Movies were drift-corrected using MotionCor2[32] via FOCUS[30]. Micrographs were processed using the FOCUS package until the 3D merging step, following standard 2D electron crystallography procedures as previously implemented in the *2dx* package[33]. This yielded the defocus, tilt geometry, lattice, and phase origin information for each micrograph. Using our newly implemented GUI, particles, i.e., patches of the 2D crystal image centered on crystal unit cells, were extracted from positions indicated by the cross-correlation profile of the classical unbending algorithm[17]. Only one lattice per image was considered, resulting in the exclusion of 76 duplicated images due to the presence of a second lattice. Unit cells with a cross-correlation (CC) peak above each micrograph's average CC peak value were picked, an approach closely coinciding with the auto-masking procedure for 2D crystals. As the unit cell of the MloK1 2D crystals had P42₁2 symmetry[29,54,57], a shift of 180 degrees along the first lattice vector was applied to the phase origin (the crystallographic unit cell) to position a MloK1 tetramer at the center of the particle box and allow the imposition of C4 symmetry in the single-particle refinement steps. The box size was of 320 square pixels roughly comprising nine MloK1 tetramers in the non-tilted views. Individual particle images were CTF-corrected by phase flipping and averaged on a per-crystal basis using new scripts in FOCUS. The particle export script also ensures that particles picked from the same 2D crystal stay in the same half-set in order to prevent inflated resolution estimations based on the FSC because of the overlap between neighboring particle boxes[21].

**Consensus refinement.** Pre-refinement using the crystal averages was performed using a custom auto-refinement script based on FREALIGN version 9.11[28] (see Suppl. Note 2 for details). Subsequent refinement after convergence was performed using the same auto-refinement procedure, but now using all particles and inheriting the alignments determined in the pre-refinement and defocus values as estimated at the center of the particle window according to the initial tilt geometry, with CTF correction performed internally by FREALIGN using Wiener filtering[25,58]. Both, in the pre-refinement and the refinement steps, a spherical mask was initially applied to the reference 3D reconstruction leaving a region comprised of about nine MloK1 tetramers for processing. To prevent reference bias, the highest resolution limit used for particle alignment was 7.52 Å.

**3D classification.** A focused spherical mask on the central MloK1 tetramer was applied to subtract the signal from the neighboring tetramers using RELION[26,42]. Afterwards, the signal-subtracted particle stack was subjected to 90 cycles of maximum-likelihood 3D classification in FREALIGN[28] using eight classes and a resolution limit of 7.0 Å. No alignments were performed at this stage. To decrease the processing time, for this classification the particle stack was downsampled to a pixel size of 2.6 Å by Fourier cropping. Asymmetric classification using both C1 symmetry and the localized reconstruction[43,44] of a single suitably masked monomer to search for deviations from C4 symmetry were also attempted.

**Map postprocessing.** For post-processing and analysis of every map, a box of 104 cubic voxels containing only the central MloK1 tetramer was extracted from the larger half-maps. The box center was translated by 12 voxels in the Z direction to coincide with the center of the MloK1 tetramer before cropping out the smaller volume. FSC curves between half-maps were calculated using a spherical mask of 42 voxels radius and a soft cosine-edge of 6 voxels width, and corrected for the relative volumes of the particle and the mask within the box[8]. The maps were sharpened by deconvolving the MTF curve of the Gatan K2 detector at 300 kV and then using the phenix.auto_sharpen program[59], and low pass-filtered at the respective resolution cutoffs defined by the 0.143 threshold criterion[39] using a soft cosine-edge filter. Resolution was also assessed using the ½-bit criterion[40] with an

estimated particle diameter of 100 Å and four-fold symmetry. A Python script called *focus.postprocess* was written based on the MRCZ module[60] and included as a command-line tool in FOCUS. Local resolution maps were calculated using Blocres[61].

**Model building.** A new model was assembled by taking the N-terminal from PDB 6EO1[29] (residues 1–6), the TMD from PDB 3BEH[41] (residues 7–219), the CNBD from PDB 3CL1[62] (residues 220–349) and the C-terminus also from PDB 6EO1[29] (residues 350–355). These domains were individually rigid-body fitted into our consensus map using UCSF Chimera[63] and then saved together as a single chain in a new PDB file. Atoms of incomplete residues were filled in using Coot[64]. The model was then flexibly fit into the consensus map using Normal Mode Analysis with iMODFIT[45] at a resolution limit of 4.0 Å. Riding hydrogens were added to prevent steric clashes in the subsequent refinement[65]. Secondary structure annotation was calculated using *ksdssp*[66] and manually adjusted in UCSF Chimera. This single chain was then refined into the map using *phenix.real_space_refine*[46] with electron scattering form factors, global minimization and B-factor refinement. Modelling issues, such as Ramachandran outliers, rotamer outliers, and steric clashes, were monitored using Molprobity[67] and manually corrected in Coot, always followed by real-space refinement rounds in PHENIX[68]. Upon convergence, three more copies of the chain were generated and rigid-body fitted into the map in UCSF Chimera to account for the tetrameric channel. This was followed again by iterations of real-space refinement in PHENIX and manual tweaking in Coot whenever necessary, which were always followed by refinement rounds in PHE-NIX. Finally, potassium ions and associated restraints were added to the model at putative positions to optimize the geometry of the selectivity filter and real-space refined again in PHENIX. Model quality metrics were assessed throughout refinement using Molprobity, EMRinger[69], and per-residue plots in Coot.

Based on the consensus model and the eight maps obtained after convergence of 3D classification in FREALIGN, we generated an ensemble of models representing the conformational variability of MloK1. The consensus model was flexibly fit into each map using Normal Mode Analysis by iMODFIT[45] followed by five cycles of real-space refinement in PHENIX with electron scattering form factors, global minimization and B-factor refinement, always using the global resolution determined for each 3D class, which ranged from 4.4 Å (class 3) to 5.6 Å (class 8). For comparison, every model in the ensemble was superposed against each other and against the consensus model using *phenix.superpose_pdbs*, and the RMSD between the C-α atoms was computed as a similarity measure. Based on the pairwise RMSD matrix resulting from the eight models, hierarchical agglomerative clustering[70] was performed using the single linkage criterion with the *scikit-learn* Python module[71]. For visual analysis, all members of the ensemble were superposed onto the consensus model based on the selectivity filter only (residues 174:180). Distances and angles were calculated using the Axes/Planes/Centroids tool[72] in UCSF Chimera.

**Data analysis.** Results were analyzed with the aid of Python scripts based on the MRCZ[60], NumPy (http://www.numpy.org), scikit-learn[71], and SciPy (http://www.scipy.org) modules.

**Figures and animations.** Plots were generated using the *matplotlib* Python module (http://www.matplotlib.org). Figures and movies were made with the aid of UCSF Chimera[63].

**Reporting summary.** Further information on experimental design is available in the Nature Research Reporting Summary linked to this article.

## Data availability
The raw data are deposited in the EMPIAR database, accession code EMPIAR-10233. The full consensus map is deposited at the EMDB, accession code EMD-4439. The cropped consensus map used for resolution estimation is deposited at the EMDB under accession code EMD-4432 and the fitted model is deposited at the PDB, accession code 6I9D. The ensemble of maps and models derived from 3D classification have been deposited to the EMDB and the PDB under the following accession codes, respectively: class 1, EMD-4441, PDB 6IAX; class 2, EMD-4513, PDB 6QCY; class 3, EMD-4514, PDB 6QCZ; class 4, EMD-4515, PDB 6QD0; class 5, EMD-4516, PDB-6QD1; class 6, EMD-4517, PDB 6QD2; class 7, EMD-4518, PDB 6QD3; class 8, EMD-4519, PDB 6QD4. Other data are available from the corresponding author upon request.

## Code availability
The source code for exporting a 2D crystal project to SPA programs is available within the FOCUS project at http://github.com/C-CINA/focus. The source code for the modified FREALIGN version is available from the repository at http://github.com/C-CINA/frealign-2dx.

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

## Acknowledgements

We thank N. Grigorieff for software support, D. Herbst and T. Maier for computational resources and discussions, C. Nimigean and R. Adaixo for discussions, and S. Müller for support in manuscript preparation. Calculations were performed at sciCORE of the University of Basel. R.D.R. acknowledges funding from the Fellowships for Excellence program sponsored by the Werner-Siemens Foundation and the University of Basel. This work was supported by the Swiss National Science Foundation, grant 205320_166164 and the NCCR TransCure.

## Author contributions

R.D.R. conceived the computational experiments, processed the data and analyzed the results. N.B. and R.D.R. wrote or modified computer programs and scripts. J.K. expressed and purified MloK1 and prepared the 2D crystal sample. M.C. and J.K. collected cryo-EM data. H.S. initiated and supervised the project. R.D.R. and H.S. wrote the manuscript, with the support from all authors.

## Additional information

**Competing interests:** The authors declare no competing interests.

