## [Peer Review File · Nature Communications]

Reviewers' comments:

Reviewer #1 (Remarks to the Author):

In this work, the author's combine the techniques of electron crystallography and single particle data processing to improve structure determination from 2D crystals. 2D crystallography is notoriously difficult, but can provide valuable information about the function of membrane proteins in a lipid environment, therefore improvements to the data processing work flow are very important. Implementation of the procedures within a user friendly program, FOCUS, is an added benefit of the presented method. The research is very good and the manuscript is well written, however there are a few minor points that should be addressed.

1)Page 6 lines 11-12: the sentence "Next, the new tab called "Particles" in the graphical user interface (GUI) allows to pick particles..." does not read right. The authors probably mean "... user interface (GUI) allows particle picking.. " or something similar.

2) Page 8 lines 2-4: "but in SPA processing these would lead to exaggerated resolution estimates due to overlap between particles picked from different lattices in the same image". Could the authors explain this a bit more, or maybe supply a reference for this statement?

3)In 2D crystallography the tilted images typically produce lower quality data. Is this also the case when this single particle based method is applied? In the final processing do the particles that come from tilted data show a trend of having lower resolution relative to untilted particles/crystals, or do these methods improve the tilted data so that it is at the same quality as the data from untilted images?

4) Supplemental figure three shows untilted classes that are highlighted with a star. Is this just to point out these classes, or is there something special with these classes relative to the tilted classes? If so, this should be made more clear.

Reviewer #2 (Remarks to the Author):

The manuscript by Righetto et al describes the structure of the bacterial MloK1 potassium channel at 4 Å resolution, determined by a combination of electron crystallography and single-particle cryo-EM of two-dimensional crystals. The single-particle processing approach yields a number of structures that differ slightly in the intracellular nucleotide-binding domain, which is considerably less well defined than the channel-forming trans-membrane domain. The slightly different structures are interpreted in terms of conformational changes in response to nucleotide binding. However, since the bound nucleotide is not resolved, this interpretation must be regarded as speculative.

The present manuscript follows another paper from the same laboratory on the cryo-EM structure of the same channel by essentially the same approach, only without the single-particle processing aspect. The resolution of the previous MloK1 map should be clearly stated in the introduction. At present, this critical information is hidden in the Results.

Another piece of critical information that cannot be found in the introduction (or indeed anywhere in the manuscript) is the molecular mass of the channel. The only information on the size of the MloK1 tetramer is gleaned indirectly from Figure Legend 2, which states that nine tetramers have a total molecular weight of more than 1 MDa, so one tetramer must be around 150 kDa. Structures of membrane proteins in this size range can now be determined by single-particle cryo-EM at significantly higher resolution, especially if they have fourfold symmetry, as the MloK1 channel. A few references to some of these recently published structures would not go amiss.

Apparently the resolution of the earlier cryo-EM structure of the MloK1 channel was 4.5 Å. The improvement to 4.0 Å in the present manuscript is significant, but slightly disappointing, given the amount of extra work invested by the authors. The factors that limit the resolution should be

discussed.

One key limiting factor must be the fact the data set does not include any side views of the channel, which is an inevitable consequence of using two-dimensional crystals. The situation resembles one in single-particle processing when adsorption of a protein to the support film results in preferential orientation, which in this case is extreme. With non-crystalline proteins, the problem can often be resolved by changing the method of specimen preparation or the support film. With two-dimensional crystals these are not viable options, and this is a serious disadvantage for structure determination.

As a result, the resolution is not only low, but also anisotropic. This may explain why even large sidechains are much less well resolved than would be expected in a 4 Å map. The Results section asserts that the resolution is isotropic to 4 Å, but the map does not look like it. In electron crystallography is usual to give the resolution in two mutually perpendicular directions, in the x-y plane and in z. What is the z resolution of the map in Figure 2, and how is it measured?

Cryo-EM structures of similarly-sized membrane proteins have been determined by single-particle cryo-EM at substantially higher resolution than 4 Å. What then is the advantage of two-dimensional crystals? Is it really the membrane environment (as suggested in the Discussion)? Purified membrane proteins can easily be (and often are) reconstituted into lipid nanodiscs for structure determination by SPA. For the benefit of the non-expert reader, these questions should be discussed in some detail.

Other comments:

Introduction: "several" high resolution structures from two-dimensional crystals presumably means more than two. Since there are not very many, non-expert readers might like to know which ones they are.

The map in Figure 2 is not an "electron density map"

Answer to Reviewer Comments.

Below are the reviewer comments in blue, and our answers in black font.

Reviewer #1 (Remarks to the Author):

In this work, the author's combine the techniques of electron crystallography and single particle data processing to improve structure determination from 2D crystals. 2D crystallography is notoriously difficult, but can provide valuable information about the function of membrane proteins in a lipid environment, therefore improvements to the data processing work flow are very important. Implementation of the procedures within a user friendly program, FOCUS, is an added benefit of the presented method. The research is very good and the manuscript is well written, however there are a few minor points that should be addressed.

We thank this reviewer for the kind words.

1) Page 6 lines 11-12: the sentence "Next, the new tab called "Particles" in the graphical user interface (GUI) allows to pick particles..." does not read right. The authors probably mean "... user interface (GUI) allows particle picking.. " or something similar.

We rephrased the sentence to:

Next, the graphical user interface (GUI) features a new tab called "Particles" in which the user can perform particle picking, i.e., window patches from the 2D crystal images (Suppl. Fig. 1), which are boxed, assembled into a particle stack, and submitted towards implemented SPA workflows, using RELION or FREALIGN within FOCUS.

2) Page 8 lines 2-4: *"but in SPA processing these would lead to exaggerated resolution estimates due to overlap between particles picked from different lattices in the same image". Could the authors explain this a bit more, or maybe supply a reference for this statement?*

We added the following clarification and a reference where the issue of multiple lattices is discussed:

"(...) i.e., it would effectively introduce duplicated particles in the dataset. This can occur for example if two 2D crystals happen to be on top of each other on the support film, or if the periodical structure under study has a three-dimensional component, such as a microtubule or vesicle, which becomes flattened during grid preparation."

3) *In 2D crystallography the tilted images typically produce lower quality data. Is this also the case when this single particle based method is applied? In the final processing do the particles that come from tilted data show a trend of having lower resolution relative to untilted particles/crystals, or do these methods improve the tilted data so that it is at the same quality as the data from untilted images?*

Yes, it is true that the tilted images are still of lower quality in comparison to the non-tilted ones. We added the following sentences in the introduction and in the conclusion, to clarify:

Introduction:

Furthermore, images acquired from tilted samples are of lower quality when compared to images of non-tilted samples, because the increased effective specimen thickness also increases the number of inelastic electron scattering events and reduces the image contrast.

Conclusion:

(...), and images of tilted samples are generally of lower quality than of non-tilted samples, (...)

4) Supplemental figure three shows untilted classes that are highlighted with a star. Is this just to point out these classes, or is there something special with these classes relative to the tilted classes? If so, this should be made more clear.

The star just indicates that these classes correspond to non-tilted classes. We updated this part of Suppl. Fig. 3 caption to read:

Images of non-tilted samples (tilt angle < 5°) are marked with a star.

Reviewer #2 (Remarks to the Author):

The manuscript by Righetto et al describes the structure of the bacterial MloK1 potassium channel at 4 Å resolution, determined by a combination of electron crystallography and single-particle cryo-EM of two-dimensional crystals. The single-particle processing approach yields a number of structures that differ slightly in the intracellular nucleotide-binding domain, which is considerably less well defined than the channel-forming trans-membrane domain. The slightly different structures are interpreted in terms of conformational changes in response to nucleotide binding. However, since the bound nucleotide is not resolved, this interpretation must be regarded as speculative.

We have now rephrased the corresponding paragraph on page 13, lines 23, as:

Following Rangl et al., a possible interpretation is that the extended “up-state” orientation of the CNBDs corresponds to the cAMP-free orientation, while the compact “down-state” orientation with the CNBDs approaching the membrane, is the cAMP-bound conformation. Such insights have been obtained previously from lower resolution structures and high-speed AFM experiments. Our data and the processing workflow presented here, now provides higher-resolution data that invite for a more detailed analysis of this hypothesis in 3D models of MloK1.

1) The present manuscript follows another paper from the same laboratory on the cryo-EM structure of the same channel by essentially the same approach, only without the single-particle processing aspect. The resolution of the previous MloK1 map should be clearly stated in the introduction. At present, this critical information is hidden in the Results.

This information was added to the Introduction:

“(…) whose structure has been previously resolved to 4.5 Å”

2) Another piece of critical information that cannot be found in the introduction (or indeed anywhere in the manuscript) is the molecular mass of the channel. The only information on the size of the MloK1 tetramer is gleaned indirectly from Figure Legend 2, which states that nine tetramers have a total molecular weight of more than 1 MDa, so one tetramer must be around 150 kDa. Structures of membrane proteins in this size range can now be determined by single-particle cryo-EM at significantly higher resolution, especially if they have fourfold symmetry, as the MloK1 channel. A few references to some of these recently published structures would not go amiss.

The molecular weight information was added to the Introduction:

“The total molecular weight of the tetramer is 160 kDa.”

References to ion channels similar to MloK1 solved by SPA at higher resolution were added:

“With the addition of detergents or reconstitution in lipid nanodiscs, ion channels similar to MloK1 in mass have been consistently resolved in the resolution range of ~3.5 Å by SPA recently (REFS).”

3) Apparently the resolution of the earlier cryo-EM structure of the MloK1 channel was 4.5 Å. The improvement to 4.0 Å in the present manuscript is significant, but slightly disappointing, given the amount of extra work invested by the authors. The factors that limit the resolution should be discussed.

We couldn't agree more, especially to the comment about the amount of extra work invested...., unfortunately ;-). The following points were added to the discussion:

The overall resolution of the map improved from 4.5 to 4.0 Å using our single particle approach. The main bottlenecks to resolution by this method are presumably the lack of side views due to the tilting limitations, and the inherent lower quality of the high tilt images. Also, in this particular MloK1 dataset the pixel size at the sample level was relatively large (1.3 Å), which means the data was collected at a range of the K2 Summit's camera detective quantum efficiency that is suboptimal from the perspective of the SNR.

One key limiting factor must be the fact the data set does not include any side views of the channel, which is an inevitable consequence of using two-dimensional crystals. The situation resembles one in single-particle processing when adsorption of a protein to the support film results in preferential orientation, which in this case is extreme. With non-crystalline proteins, the problem can often be resolved by changing the method of specimen preparation or the support film. With two-dimensional crystals these are not viable options, and this is a serious disadvantage for structure determination.

4) As a result, the resolution is not only low, but also anisotropic. This may explain why even large sidechains are much less well resolved than would be expected in a 4 Å map. The Results section asserts that the resolution is isotropic to 4 Å, but the map does not look like it. In electron crystallography is usual to give the resolution in two mutually perpendicular directions, in the x-y plane and in z. What is the z resolution of the map in Figure 2, and how is it measured?

We added the following clarification to the Results section:

This 4.0 Å resolution estimate based on the FSC corresponds to an isotropically averaged measurement. The map's resolution is better in the xy membrane plane than it is along the z axis. By calculating the Fourier ring correlation between the central xy slices (z=0) of the unmasked half-maps we estimate

resolution to be slightly better than 4.0 Å in this plane, whereas in the orthogonal xz plane (y=0), which is identical to the yz plane due to the imposed symmetry, the estimated resolution is about 4.7 Å considering the FSC 0.143 threshold (Suppl. Fig. 4f). These two orthogonal resolution estimates are consistent with the observed features of the consensus map.

5) Cryo-EM structures of similarly-sized membrane proteins have been determined by single-particle cryo-EM at substantially higher resolution than 4 Å. What then is the advantage of two-dimensional crystals? Is it really the membrane environment (as suggested in the Discussion)? Purified membrane proteins can easily be (and often are) reconstituted into lipid nanodiscs for structure determination by SPA. For the benefit of the non-expert reader, these questions should be discussed in some detail.

We agree. In many cases, going for 2D crystals seems to be a really bad idea nowadays. Nevertheless, there do exist cases, where a structural analysis of a 2D crystalline state is required. We had listed several scenarios, where 2D crystal studies might be of interest, in the introduction. For clarity, we also added the following points to the discussion, in connection to the above comment about membrane proteins being solved by SPA at higher resolution:

With the addition of detergents or reconstitution in lipid nanodiscs, ion channels similar to MloK1 in mass have been consistently resolved in the resolution range of ~3.5 Å by SPA recently. Nevertheless, cases exist where the structure of a membrane protein in the lipid environment provided by a 2D crystal is important, such as when the crystals occur naturally in the cell membrane, or when a 2D periodic array is artificially designed.

Other comments:

6) Introduction: "several" high resolution structures from two-dimensional crystals presumably means more than two. Since there are not very many, non-expert readers might like to know which ones they are.

Additional references were included.

7) The map in Figure 2 is not an "electron density map"

The title of the Figure caption was changed to read:

"The cryo-EM map of full-length MloK1 at 4 Å."

General Comments:

=====
FOCUS software system:
=====

The FOCUS software package is hosted as open-source on GITHUB and can be obtained with the command:
git clone <https://github.com/C-CINA/focus>

The source code for the modified FREALIGN version is available in the following GitHub repo:
<https://github.com/C-CINA/frealign-2dx>

Installation instructions are detailed at
<https://focus.c-cina.unibas.ch/wiki/doku.php>

The Software requires additional 3rd-party softwares, as described at
https://focus.c-cina.unibas.ch/wiki/doku.php?id=1_0:external-tools

The Software runs on LINUX distributions, such as UBUNTU, RedHat, or others.
A strong GPU is recommended, such as an NVIDIA GTX1080 or similar.

Typical install time on a strong LINUX computer for FOCUS is 1 hours. For additional packages, install time depends mostly on the internet connection speed.

Instructions to run on data are described in the manuscript and on focus-em.org.
The expected output is described in the manuscript.

The expected run time for a demo on a powerful linux computer (24 cores, 128GB RAM, 1xGTX1080 GPU, 100TB Harddrive) is several days.

Detailed instructions for use are available on focus-em.org, and in the manuscript.

=====
Test data:
=====

A test dataset to reproduce the work in this manuscript is available at
<https://www.ebi.ac.uk/pdbe/emdb/empiar/entry/10006/>

The raw data and intermediate processing data from this work (aligned averages, particle stacks, particle coordinates and alignment metadata) have been deposited under EMPIAR-10233, and will be available under
<https://www.ebi.ac.uk/pdbe/emdb/empiar/entry/10233/>

=====
Maps (EMD) and atomic structure coordinates (PDB):
=====

Consensus map (central tetramer only)
EMD-4432

PDB-6I9D

Consensus map (full reconstruction 2x2 unit cells)
EMD-4439

Class 1
EMD-4441
PDB-6IAX

Class 2
EMD-4513
PDB-6QCY

Class 3
EMD-4514
PDB-6QCZ

Class 4
EMD-4515
PDB-6QD0

Class 5
EMD-4516
PDB-6QD1

Class 6
EMD-4517
PDB-6QD2

Class 7
EMD-4518
PDB-6QD3

Class 8
EMD-4519
PDB-6QD4